# 2D NMR Analysis as a Sensitive Tool for Evaluating the Higher-Order Structural Integrity of Monoclonal Antibody against COVID-19

**DOI:** 10.3390/pharmaceutics14101981

**Published:** 2022-09-20

**Authors:** Francesca Cantini, Emanuele Andreano, Ida Paciello, Veronica Ghini, Francesco Berti, Rino Rappuoli, Lucia Banci

**Affiliations:** 1Magnetic Resonance Center (CERM), Consorzio Interuniversitario Risonanze Magnetiche Metallo Proteine (CIRMMP), Department of Chemistry “Ugo Schiff”, University of Florence, 50019 Sesto Fiorentino, Italy; 2Monoclonal Antibody Discovery (MAD) Lab, Fondazione Toscana Life Sciences, 53100 Siena, Italy; 3GSK Vaccines, Technical R&D, 53100 Siena, Italy; 4Department of Biotechnology, Chemistry and Pharmacy, University of Siena, 53100 Siena, Italy

**Keywords:** monoclonal antibodies, nuclear magnetic resonance, higher order structure, COVID-19

## Abstract

The higher-order structure (HOS) of protein therapeutics has been confirmed as a critical quality parameter. In this study, we compared 2D ^1^H-^13^C ALSOFAST-HMQC NMR spectra with immunochemical ELISA-based analysis to evaluate their sensitivity in assessing the HOS of a potent human monoclonal antibody (mAb) for the treatment of coronavirus disease 2019 (COVID-19). The study confirmed that the methyl region of the 2D ^1^H-^13^C NMR spectrum is sensitive to changes in the secondary and tertiary structure of the mAb, more than ELISA immunoassay. Because of its highly detailed level of characterization (i.e., many ^1^H-^13^C cross-peaks are used for statistical comparability), the NMR technique also provided a more informative outcome for the product characterization of biopharmaceuticals. This NMR approach represents a powerful tool in assessing the overall higher-order structural integrity of mAb as an alternative to conventional immunoassays.

## 1. Introduction

A potent Fc-engineered human monoclonal antibody (mAb), named MAD0004J08, neutralizes in vitro multiple severe acute respiratory syndrome coronavirus 2 (SARS-CoV-2) variants of concern, such as alpha, beta, gamma, and delta, and the omicron variant, though with reduced potency [1,2]. Data collected in a Phase 1 clinical study (EudraCT N.: 2020-005469-15 and ClinicalTrials.gov Identifier: NCT04932850) confirmed that a single-dose administration of MAD0004J08 via the intramuscular (i.m.) route was safe and well-tolerated. It resulted in rapid serum distribution and sera-neutralizing titers higher than those of COVID-19 convalescent and vaccinated subjects, and was also sufficient to neutralize major SARS-CoV-2 variants of concern (alpha, beta, gamma, and delta) [3].

The higher-order structure (HOS) of biological macromolecules (i.e., proteins, mAbs, etc.) represents a critical quality parameter directly related to structural integrity and, consequently, to function. HOS depends on the protein’s secondary, tertiary, and quaternary structure. The characterization of HOS has been conducted so far by using biophysical methods, for instance, Fourier transform infrared spectroscopy (FTIR), circular dichroism (CD), fluorescence spectroscopy (FLD), differential scanning calorimetry (DSC), and by other immunochemical methodologies [4,5,6].

In addition, as is the case for measuring vaccine potency [7], newer analytical tools, including biochemical, cell-based, and immunochemical methods, can be applied to assess the structural integrity and functionality of mAbs.

The use of nuclear magnetic resonance (NMR) has been suggested as a technology with the potential to more accurately assess the differences between HOS and other well-established methods [8,9,10,11,12].

In this study, we compared the sensitivity of 2D ^1^H-^13^C ALSOFAST-HMQC NMR spectra with that of immunochemical ELISA-based analysis in assessing the HOS of MAD0004J08 mAb. NMR characterization was found to be a powerful tool for assessing the overall higher-order structural integrity of mAb as an alternative to the immunoassay.

## 2. Materials and Methods

### 2.1. Samples

SARS-CoV-2 mAbs batches 19G, 20G, and 33G were obtained from Menarini Biotech, Analytical Development, Latina, Italy. The analytical samples were in 20 mM phosphate and 150 mM sodium chloride buffer at pH 7.0, with a protein concentration of approximately 43 mg/mL.

To determine the assays’ sensitivity, aliquots of batches 33G and 20G were incubated in 1% H_2_O_2_ at pH 10 (by addition of NaOH) for 24 and 48 h. No quenching step was applied following hydrogen peroxide treatment. In the case of pH 10 treatment, the samples were returned to pH to 7.0 before acquiring the NMR experiments.

### 2.2. Nuclear Magnetic Resonance

^1^H-^13^C ALSOFAST-heteronuclear multiple quantum coherence (HMQC) correlation [13] spectra were collected at 700, 900, and 1200 MHz on Bruker Avance spectrometers equipped with triple-resonance, cryogenically cooled probes with *z*-axis-gradient systems. The pulse program “afhmqcgpphsf” in the Bruker library was used.

NMR analytical samples were prepared in 5 mm tubes (Wilmad LabGlass, Vineland, NJ, USA) by inserting 550 µL with 10% D_2_O (Sigma Aldrich, Merck KGaA, Darmstadt, Germany).

Spectra were collected at 40 °C for about 14 h without the application of the NUS scheme. ^1^H-^13^C ALSOFAST-HMQC correlation spectra were repeated at different times on samples that had just thawed, after seven days, after thirty days, and after sixty days. Between each experiment, the samples were maintained in the NMR tubes and stored at 4 °C. ^1^H-^13^C ALSOFAST-HMQC correlation spectra were recorded at 700 MHz with 1.024 scans and 256 × 900 complex points, corresponding to spectral widths of 29.8 × 19.8 ppm with acquisition times of 24.4 and 32 ms in the t_1_ (^13^C) and t_2_ (^1^H) domains, respectively. At 900 MHz, the ^1^H-^13^C ALSOFAST-HMQC correlation spectra were recorded with 1.024 scans and 256 × 1.152 complex points. Acquisition times of 18.9 and 32 ms in the t_1_ (^13^C) and t_2_ (^1^H) domains were set, respectively. At 1.2 GHz, the spectra were recorded with 1280 scans and 256 × 1.536 complex points. Acquisition times of 14.1 and 32 ms in the t_1_ (^13^C) and t_2_ (^1^H) domains were set, respectively.

The ^1^H and ^13^C carriers were placed on water resonance and at 20 ppm. A recycle delay of 0.25 s was applied. Reburp selective pulse [14], on resonance on methyl groups with a bandwidth of 29.0 ppm, was used for selective refocusing of methyl resonances. Data were processed with Topspin 4.07. Apodized with a qsine function and ssb values of 3 and 2 in F_2_ and F_1_ dimensions, respectively, were used. Forward linear prediction in the F1 dimension was applied prior to Fourier transform.

### 2.3. Statistics on NMR Data

All statistical analyses were applied on binned 2D NMR spectra through R software (R 4.1.2, R Foundation for Statistical Computing, Vienna, Austria) using in-house scripts. To this end, each spectrum in the region of −1.0 to 2.8 ppm (^1^H dimension) and 8.0 to 26.0 ppm (^13^C dimension) was segmented into 0.01 and 0.1 ppm chemical shift bins, for the ^1^H and ^13^C dimensions, respectively. Prior to all statistical analyses, the binned data were normalized using total area normalization.

The corresponding spectral areas were integrated using AssureNMR software (Bruker BioSpin GmbH—Rheinstetten, Germany). Unsupervised principal component analysis (PCA) was used in exploratory analysis to obtain an overview of the data (visualization in a reduced space and presence of clusters). The distances among the different spectra were calculated using Euclidean distances.

^1^H-^13^C ALSOFAST-HMQC spectra were analyzed over the bounds of 8 to 26.0 ppm in F_1_ and −1 to 2.8 ppm in F_2_. Peak picking was performed manually using CARA software [15]. The combined chemical shift difference (CCSD) was defined as:$$\mathrm{CCSD}=\sqrt{\frac{1}{2}\left[{\left(\delta_{H}-\delta_{Href}\right)}^{2}+{\left(0.251\delta_{C}-0.251\delta_{Cref}\right)}^{2}\right]}$$
where $\delta_{H}\;\mathrm{and}\;\delta_{C\;}$ are, respectively, the ^1^H and ^13^C chemical shifts of the analyzed cross-peak; $\delta_{H,ref}\;\mathrm{and}\;\delta_{C,ref\;}$ are the ^1^H and ^13^C chemical shifts, respectively, for the same peak of the batch 20G taken as the reference spectrum. The error bars of the average CCSD values were calculated using the standard error of the mean value as described in [16].
$$\mathrm{Error}=\mathrm{average}(\mathrm{CCSD})\;\pm \;1.96\;\times \;\frac{\sigma }{\sqrt{n}}$$
where σ is the sample standard deviation, and *n* is the number of analyzed spectra.

### 2.4. ELISA-Based Assay

An amount of 3 μg/mL of streptavidin (Thermo Fisher) was diluted in carbonate-bicarbonate buffer (E107, Bethyl Laboratories, Montgomery, TX, USA) and used to coat 384-well plates (microplate clear, Greiner Bio-one, Monroe, NC, USA). Coated plates were incubated overnight at room temperature. The following day, streptavidin-coated plates were incubated with 3 μg/mL of SARS-CoV-2 S protein diluted in PBS at pH 7.2 for 1 h at room temperature. Then, a blocking step was performed by adding 50 µL of 1% bovine serum albumin (BSA; Sigma, Merck KGaA, Darmstadt, Germany) in PBS and incubating the plates for 1 h at 37 °C. After blocking, mAbs were diluted in sample buffer (PBS, 1% BSA and 0.05% Tween-20) and tested in a final volume of 25 µL/well (starting concentration 10.000 ng/mL; step dilutions 1/2). The plates were then incubated for 1 h at 37 °C. Next, 25 µL/well of secondary antibody coupled to alkaline phosphatase diluted 1:2000 in sample buffer was added, and the plates were incubated for 1 h at 37 °C. To detect mAbs binding to SARS-CoV-2 S protein, 25 µL/well of para-nitrophenyl phosphate (p-NPP; Sigma, Merck KGaA, Darmstadt, Germany) was used, and the reaction was measured at a wavelength of 405 nm. Plates were washed three times with 100 μL/well of washing buffer (PBS and 0.05% Tween-20; Sigma, Merck KGaA, Darmstadt, Germany) after each incubation step. Sample buffer was used as a blank, and results were considered positive if OD at 405 nm (OD405) was twice that of the blank.

## 3. Results

### 3.1. Comparison of Different mAb Batches and Stability over Time by NMR Technique

The ^1^H-^13^C ALSOFAST-HMQC NMR spectra, acquired at different times, were compared to analyze the stability of the mAb over time and the equivalence of different batches of the mAb, i.e., 19G and 20G. NMR spectra were acquired at four different time points, i.e., just after thawing (T0), after seven days (T7), after 30 days (T30), and after 60 days (T60), with three NMR spectrometers operating at different magnetic field strengths (700, 900, and 1200 MHz). Figure 1A–F show the ^1^H-^13^C methyl correlation spectra acquired correspondingly on the 19G and 20G mAb samples at the three magnetic fields.

Using spectra recorded at 700 MHz on batch 20G as the reference for peak picking of all spectra, a peak list with approximately 110 cross-peaks was manually defined and used for all other acquired ^1^H-^13^C spectra to allow an in-depth comparative analysis. Average combined chemical shift difference (average CCSD) analysis was used to evaluate whether the NMR peaks of the 700 MHz spectra acquired on batches 19G and 20G over time (Table 1) showed spectral differences. Overall, high mAb stability over time was demonstrated by the minimal frequency changes of the cross-peaks in the 2D NMR experiments, and, consequently, by the low average CCSD values.

Taken together, these data indicate that both batches, 19G and 20G, maintained their 3D structure, even in their minor, local details over time. The CCSD analysis (Figure 2) also indicated that batch-to-batch variability was small, as the average CCSD value was 5.2 ± 2.5 ppb at T0. Our CCSD analysis is consistent with the NMR data already reported on monoclonal antibodies to assess their higher order structures [10,17].

To further identify HOS perturbations over time in the two analyzed mAb production batches, 19G and 20G, the ^1^H-^13^C methyl correlation spectra were chemometrically compared using principal component analysis (PCA) and Euclidean distance [9]. The results of PCA (Figure 3) highlighted that it was not possible to distinguish and cluster the two 19G and 20G mAbs samples, confirming high reproducibility in the mAbs preparation. Moreover, the PCA score plots in Figure 3B–D show that the spectra acquired at different time points did not separately cluster, indicating good long-term stability.

The Euclidean distance was also used to evaluate the similarity between the NMR spectra of the 19G and 20G batches. The distance values between the spectra of each antibody batch at T0 and their corresponding spectra acquired at three time points (T7, T30, and T60) did not significantly change, with the average value being 0.32, for both batch 19G and 20G, after T0 vs. T7 and T0 vs. T30 comparisons. Moreover, the average distances calculated between the T0 spectra of 19G and 20G were in the same order (Figure 4). Therefore, the results of chemometric analysis confirmed that the two production batches, 19G and 20G, were comparable and that HOS analysis of these batches indicated they were stable over 60 days.

### 3.2. Stress Testing: Evaluation of HOS Perturbations after Incubation in H_2_O_2_ and at pH 10 by NMR and ELISA Methods

Stress testing of the mAbs was performed to identify possible changes in structural stability upon the addition of H_2_O_2_ or increase in pH.

The same ^1^H-^13^C ALSOFAST-HMQC spectra, as described above and used for assessing the structural stability over time and different production batches, and an immunochemical method (enzyme-linked immunosorbent assay (ELISA)) to measure the binding affinity of mAb with SARS-CoV-2 spike (S) protein, were used to compare the untreated with the stressed materials. ELISAs are commonly used to measure the potency of biological products and evaluate their stability.

The ^1^H-^13^C methyl correlation spectra were acquired at 1200 MHz on mAb samples (20G and 33G batches) that had been incubated with 1% of H_2_O_2_ or at pH 10 for 24 and 48 h. The spectra were compared with those of the two reference batches, 19G and 20G, collected in the same conditions.

The ^1^H-^13^C NMR spectra of both the H_2_O_2_-treated and the high-pH-exposed samples showed significant spectral changes in chemical shifts for several methyl peaks with respect to the untreated samples (Figure 5A,B). Interestingly, the H_2_O_2_-treated samples also appeared to be slightly opalescent after the acquisition of the spectra at 313 K. ^1^H-^13^C ALSOFAST-HMQC NMR spectra comparison provided a CCSD of 12.1 ± 1.1 ppb and 14.0± 1.5 ppb between reference batch 20G and mAbs incubated with 1% of H_2_O_2_ for 24 and 48 h, respectively (Figure 6A,B). A large number of NMR signals disappeared in the ^1^H-^13^C ALSOFAST-HMQC spectra acquired on the H_2_O_2_-treated samples (Figure 6A,B). Conversely, no methyl signals disappeared and smaller spectral changes (CCSD value of 8.0 ± 1.5 ppb) were observed between the reference batch 20G and the same sample incubated at pH 10 (Figure 6C). Accordingly, the values of the Euclidean distance between the reference batch 20G and the mAbs incubated with H_2_O_2_ were higher with regard to the same distances obtained for the comparison with mAbs treated at pH 10 (Figure 5C). The PCA score plot reported in Figure 5D shows that the different groups of samples, i.e., nontreated, incubated at pH10 and with H_2_O_2_ mAbs, cluster in different areas, confirming that the groups of samples had clearly different profiles.

As a next step, mAbs batch 20G was tested for its ability to bind the SARS-CoV-2 S protein trimer in its prefusion conformation by enzyme-linked immunosorbent assay (ELISA). In detail, ELISA was performed testing the mAb batch 20G before and after incubation in H_2_O_2_ and at pH 10 for 24 and 48 h. As reported in Figure 6, H_2_O_2_ and the high pH treatment did not affect mAb ability to bind the SARS-CoV-2 S protein. In fact, 20G showed the same binding profile before and after treatment.

In our previous study, we identified the epitope targeted by J08 on the SARS-CoV-2 S protein RBD [2]. As shown in Figure 7B, the J08 footprint is distant from exposed residues, which are susceptible to oxidation from H_2_O_2_ treatment. Indeed, none of the RBD histidines and tryptophans were found within the J08 footprint, while only two out of five cysteines were positioned within this region. This analysis suggests that while some binding alterations may have occurred in the H_2_O_2_-treated samples, the ELISA approach may not have been sensitive enough to detect them (Figure 7B).

## 4. Discussion

In this study, we compared the ability of 2D NMR spectra and an ELISA-based test to measure the HOS of a mAb that binds SARS-CoV-2 S.

For mAbs and biological products, the evaluation of the structural integrity and lot-to-lot comparability by applying a physicochemical approach (i.e., NMR) plays an important role and might have greater use in the future. The 2D NMR spectra provided structural information about the entire molecule, and was quick to perform, sensitive and highly reproducible.

For this mAb, method applicability at different magnetic field strengths was confirmed by the spectra acquired at 700, 900 and 1200 MHz (Figure 1). The 700 MHz spectra provided a peak list of approximately 110 cross-peaks, which were used for average CCSD analysis and to compare two mAb batches (Figure 2). High batch-to-batch comparability was confirmed, and high mAb stability over time was demonstrated.

Furthermore, in comparison with ELISA, the 2D NMR spectra were more sensitive. Although ELISA did not reveal differences in the binding affinity of mAb batches incubated in 1% H_2_O_2_ for 24 and 48 h (Figure 7), the 2D NMR technique showed significant spectral changes in chemical shifts for several methyl peaks (Figure 5 and Figure 6).

The high sensitivity of NMR to changes in the HOS may make it particularly useful for monitoring stability. Further evaluation to establish the correlation between 2D NMR results and virus neutralization may expand the applicability of this technique by determining whether early changes detected by 2D NMR and not by ELISA impact mAb immune function. For the reasons stated above, 2D NMR characterization may become a replacement in many cases for this type of HOS assessment and the product characterization of biopharmaceuticals.

## 5. Conclusions

The results confirmed that 2D ^1^H-^13^C HSQC NMR is a more sensitive tool for mAb structure than a classical ELISA immunoassay for assessing product stability. Overall, our findings confirmed the superiority of this NMR application in the assessment of HOS attributes of this mAb. Further investigation would be useful to assess whether the higher sensitivity of 2D NMR detects changes relevant for mAb immune functionality that are not detected by ELISA.

## 6. Patents

E.A., I.P., and R.R. are listed as inventors of full-length human monoclonal antibodies described in Italian patent application nos. 102020000015754 filed on 30 June 2020, 102020000018955 filed on 3 August 2020, and 102020000029969 filed on 4 December 2020 and the international patent system number PCT/IB2021/055755 filed on 28 June 2021. All patents were submitted by Fondazione Toscana Life Sciences, Siena, Italy.

## Figures and Tables

**Figure 1 pharmaceutics-14-01981-f001:**
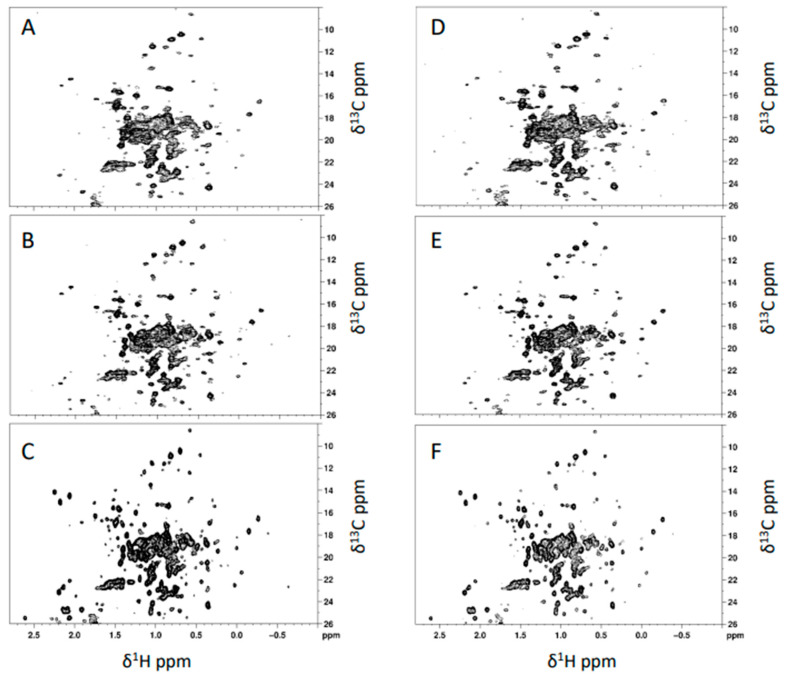
^1^H-^13^C ALSOFAST-HMQC spectra of batches 19G and 20G acquired with 700 (**A**,**B**), 900 (**C**,**D**), and 1200 MHz (**E**,**F**) spectrometers.

**Figure 2 pharmaceutics-14-01981-f002:**
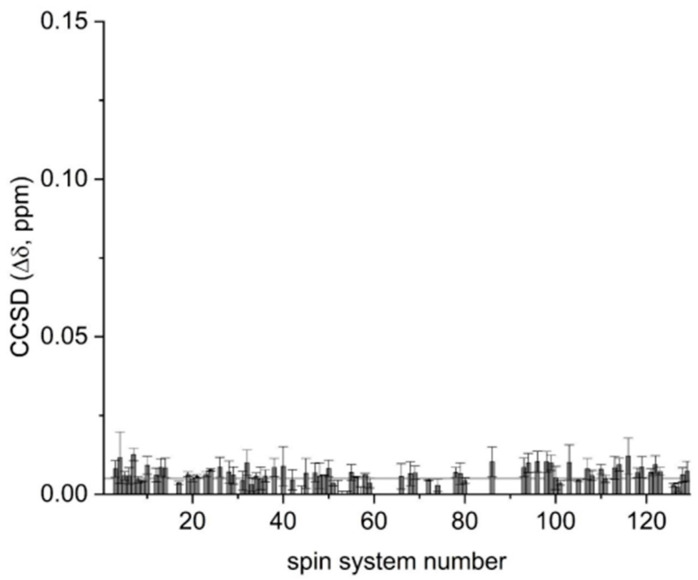
Pairwise combined chemical shift difference (CCSD) analysis. CCSD plotted along the spin systems detected, calculated between batches 19G and 20G at T0.

**Figure 3 pharmaceutics-14-01981-f003:**
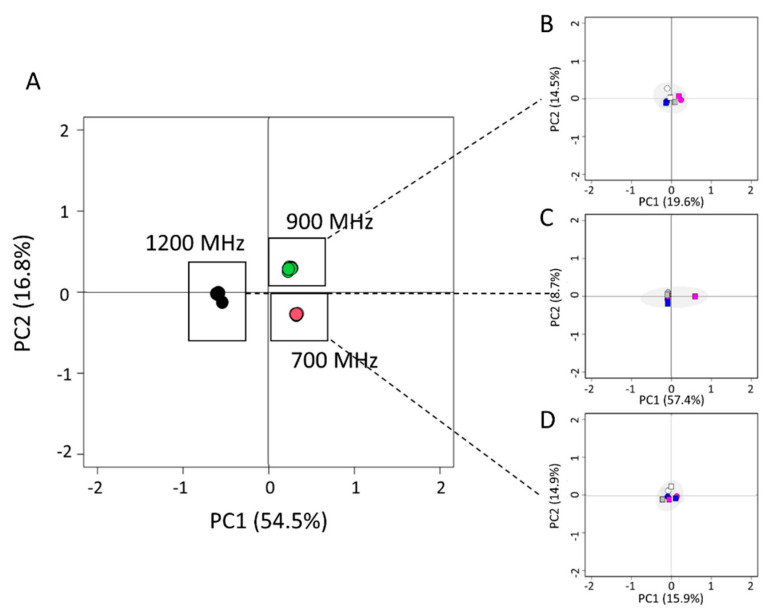
PCA score plot of ^1^H-^13^C methyl correlation spectra acquired on batches 19G and 20G at 700 (red dots), 900 (green dots), and 1200 MHz (black dots) over time (**A**). PCA score plots of batch 19G (squares) and 20G (circles) NMR spectra acquired at 900 (**B**), 1200 (**C**), and 700 MHz (**D**) over time: just thawed (white), after seven days (gray), after 30 days (blue), and after 60 days (magenta).

**Figure 4 pharmaceutics-14-01981-f004:**
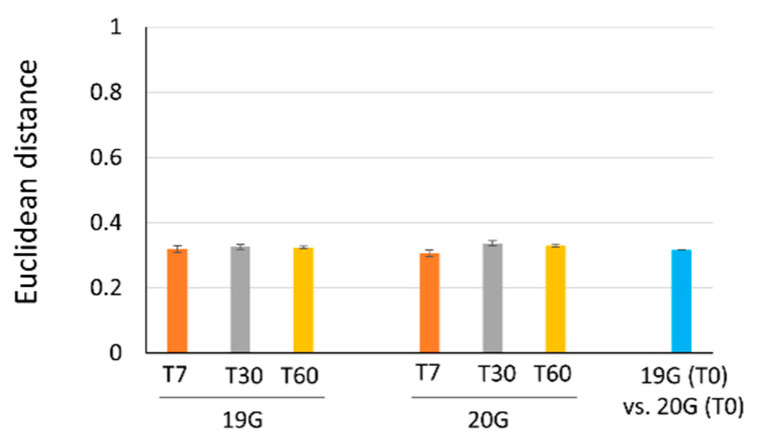
Bar plots of the Euclidean distances of batch 19G and 20G NMR spectra acquired at 700 MHz over time; the distances were calculated (i) between the spectra of both 19G and 20G obtained at each time point and the respective spectrum at T0 (T7 vs. T0: orange bars; T30 vs. T0: gray bars; T60 vs. T0: yellow bars) and (ii) between the T0 spectra of 19G and 20G (blue bars). A larger distance value indicates low similarity between the compared spectra.

**Figure 5 pharmaceutics-14-01981-f005:**
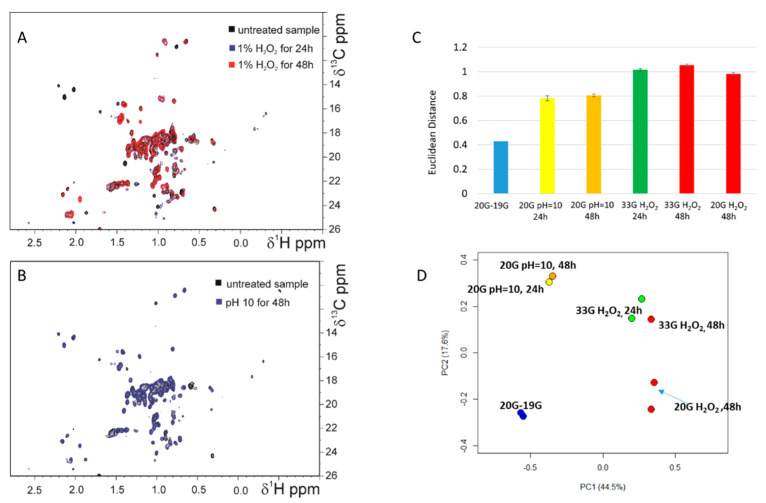
Overlay between ^1^H-^13^C ALSOFAST-HMQC spectra acquired at 1200 MHz on batch 20G and sample incubated with 1% of H_2_O_2_ (**A**); and between ^1^H-^13^C methyl correlation spectra acquired at 1200 MHz on batch 20G and sample incubated at pH 10 for 48 h (**B**). Bar plots of the Euclidean distances of batch 20G and 20G NMR spectra acquired at 1200 MHz (**C**); the distances were calculated between the spectra of 19G, 20G nontreated and 33G treated with H_2_O_2_ for 24 and 48 h (green and red bars, respectively) and 33G treated at pH 10 (yellow bar). PCA score plot of ^1^H-^13^C methyl correlation spectra acquired at 1200 MHz (**D**). References batches 19G and 20G (cyan dots), mAbs samples incubated with 1% of H_2_O_2_ for 24 h (green dots); for 48 h (red dots) and incubated at pH 10 for 48 h (yellow dots).

**Figure 6 pharmaceutics-14-01981-f006:**
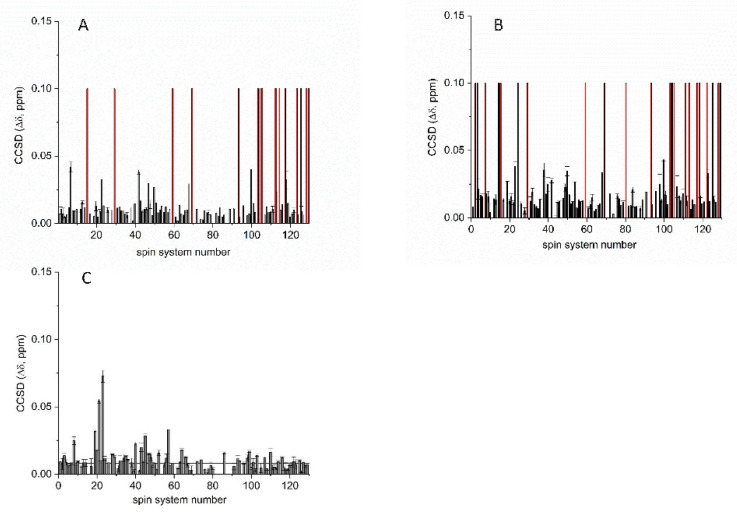
Combined chemical shift difference (CCSD) analysis. CCSD plotted along the spin system detected, calculated between batch 20G and sample incubated for 24 h with 1% of H_2_O_2_ (**A**); between batch 20G and sample incubated for 48 h with 1% of H_2_O_2_ (**B**); between batch 20G and sample incubated at pH 10 for 48 h (**C**). The lines represent the average CCSD values. The resonance positions from the ^1^H-^13^C ALSOFAST-HMQC spectrum acquired on batch 20G were used as a reference. The red bars represent the methyl cross-peak of the spin systems that disappeared after H_2_O_2_-treatment.

**Figure 7 pharmaceutics-14-01981-f007:**
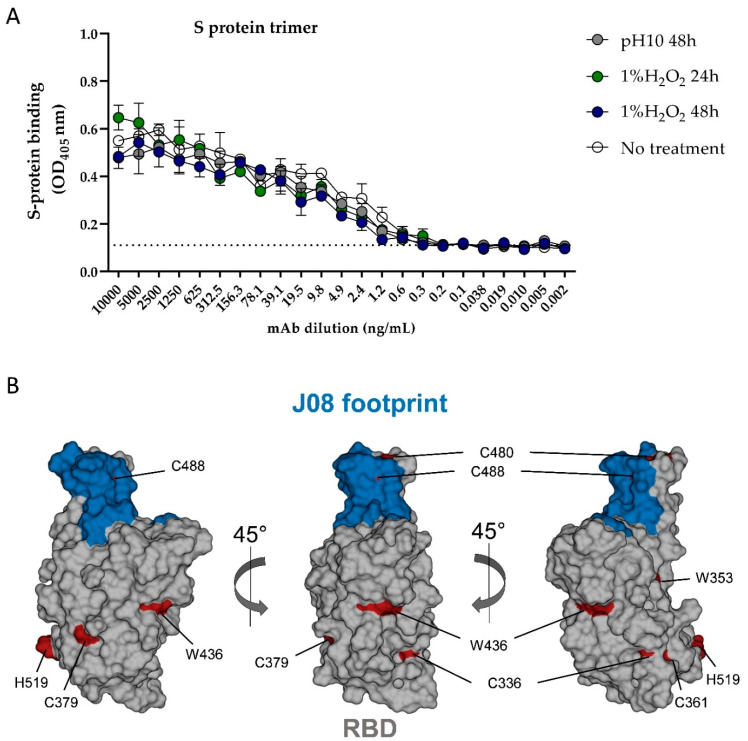
Binding measurements of mAbs batch 20G before and after incubation with 1% of H_2_O_2_ and at pH 10 for 24 and 48 h. Threshold of positivity was set as two times the value of the blank (dotted line) (**A**). Representation of the SARS-CoV-2 S protein RBD. J08 footprint is highlighted in blue; cysteines, histidines, and tryptophans are highlighted in red (**B**).

**Table 1 pharmaceutics-14-01981-t001:** Average combined chemical shift differences.

**Average CCSD (ppb)**	**mAb 19G T0**	**mAb 19G T7**	**mAb 19G T30**	**mAb 19G T60**
mAb 19G T0	0.00	2.36	3.91	4.96
mAb 19G T7	2.36	0.00	3.18	5.60
mAb 19G T30	3.91	3.18	0.00	6.52
mAb 19G T60	4.96	5.60	6.52	0.00
**Average CCSD (ppb)**	**mAb 20G T0**	**mAb 20G T7**	**mAb 20G T30**	**mAb 20G T60**
mAb 20G T0	0.00	2.38	2.08	1.91
mAb 20G T7	2.38	0.00	2.16	2.61
mAb 20G T30	2.08	2.16	0.00	1.71
mAb 20G T60	1.91	2.61	1.71	0.00

## Data Availability

Not applicable.

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
