# Peer review of "2D NMR Analysis as a Sensitive Tool for Evaluating the Higher-Order Structural Integrity of Monoclonal Antibody against COVID-19"

_pharmaceutics, 2022, doi:10.3390/pharmaceutics14101981_

Round 1

Author Response

See file enclosed

Reviewer 2 Report

This manuscript introduced a method to use 2D NMR analysis for evaluating the higher order structural integrity of monoclonal antibodies. It is well writing. I have a basic question for this study.

The manuscript shown 2D 1H-13C HSQC NMR is better than Elisa for assessing product stability. However, Elisa only shows the affinity (or avidity). It usually used for function evaluation. This study is based on structure analyze, there are several methods (structurally analyze) can be used to detect the stability of antibodies such as size exclusion, circular dichroism, Mass spectroscopy and DLS. Why you chose Elisa instead of other widely used methods to do the comparation? Thanks!

Author Response

See file enclosed

Round 2

Reviewer 1 Report

The authors adequately addressed my previous comments.

Reviewer 2 Report

Agree with the other reviewer. Thanks!